# The impact of city-county merger on firm-level total factor productivity—Evidence from China

**Sisi Yin**[1]*, **Zuyi Shen**[1], **Yunshu Gao**[2]

**1** Business School, Chengdu University of Technology, Chengdu, Sichuan, China, **2** School of Economics, Southwestern University of Finance and Economics, Chengdu, Sichuan, China

* yinwaiwen@126.com

**Data Availability Statement:** All files regarding the data used in this manuscript are available from the following link: https://doi.org/10.17605/OSF.IO/Z3PYF.

## Abstract

This study examines the potential negative effects of city-county mergers on firm-level total factor productivity (TFP). A theoretical framework is established to explain how city-county mergers could lead to higher institutional and non-institutional costs, thus influencing government decision-making and resource allocation. As a result, firms operating in the economy will absorb these costs and adjust their behavior accordingly. By employing Chinese firm-level data in a difference-in-differences approach, this study finds a statistically significant negative effect of transitioning county-level cities into municipal districts (TCD) on firm-level productivity. Additionally, this study identifies the financial situation of reformed governments and the internal environment of firms as two mechanisms through which the TCD reform influences firm-level productivity.

## 1. Introduction

In recent decades, there has been a growing trend in many countries to carry out municipal amalgamation reforms [1], allowing the government to re-draw municipal boundaries and reconfigure the administrative structures. These reforms are seen as a way to increase the coordination and effectiveness of local governments in promoting economic development. A large body of literature studies the impact of municipal amalgamation in developed countries [2–8]. However, scholars have taken divergent stances on the impact of municipal amalgamations. Proponents argue that amalgamation can lead to cost savings and economies of scale [3, 4], while opponents contend that the impact on local public expenditures and taxation is minimal and that a centralized local government structure may lead to efficiency losses due to differing preferences among local governments [5]. In other words, despite the advantages, the process of amalgamation may lead to considerable transaction costs. Consequently, firms operating within the economy will internalize these costs and adjust their behavior accordingly. The existing research primarily focuses on the macro-level impact of amalgamation, with limited attention on how amalgamation affects micro-level firms. This paper complements that literature by looking at the impact of municipal amalgamation on firm performance in China.

**Funding:** This research was supported by the National Natural Science Foundation of China (Grant No. 72303187); the Sichuan Science and Technology Program (Grant Nos. 23RKX0359 & 24NSFSC6313); and the 2022 Chengdu Philosophy and Social Sciences Planning Youth Project (Grant No. 2022C08). The funders had no role in study design, data collection and analysis, decision to publish, or preparation of the manuscript.

**Competing interests:** The authors have declared that no competing interests exist.

More specifically, we investigate whether the municipal amalgamation policy affects total factor productivity (TFP) using Chinses firm-level data in a difference-in-differences setting.

Among the developing countries that have implemented the amalgamation reforms, China offers an intriguing case for studying the impact of amalgamation on firm performance. To illustrate, the fundamental character and physical structure of China's urban spatiality is markedly distinct from that of any North American or European nation due to China's urban administrative spatial system possessing a more rigid hierarchical structure, being more rank-based and rank-conscious, and exerting stronger influence on local development. Additionally, the administrative rank of a city plays a crucial role in determining its capacity to function as an economic entity, has a profound impact on local events, and shapes the city's relationships across different scales [9]. Thus, in countries with multi-tier political hierarchies like China, assessing the impact of power shifting among different levels of government during the amalgamation process on firm performance is a pressing and unanswered question.

Specifically, after 1998, China shifted its focus on implementing city-county merger (also known as *che xian she qu*), wherein counties or county-level cities are transitioned as municipal districts of the corresponding higher-level prefecture cities. In other words, the city-county merger removed administrative barriers between central cities and counties, with counties expected to be subordinate to the prefecture-level cities [10]. This process resulted in the centralization of some administrative and financial powers of reformed county-level governments to the prefecture-level city governments. Consequently, the authority of the governments in the newly established districts decreased compared to the pre-reformed counties or county-level cities, leading to changes in decision-making related to local infrastructure investment and construction. The shift in decision-making and allocation of resources due to the city-county merger can influence firm performance, leading to variations in productivity.

In this study, we construct a simple theoretical model to rationalize that the shrinking of authority within local government results in reduced efficiency of the transactions, ultimately leading to a decline in firm-level productivity. Subsequently, the policy of transitioning county-level cities into municipal districts (TCD) is managed as a quasi-natural experiment to investigate the impact of power shifts between prefecture and county-level governments on firm-level total factor productivity (TFP) in China.

Moreover, the process of city-county merger involves turning regular counties into municipal districts and turning county-level cities into municipal districts. Most existing literature did not differentiate the above two situations and empirically examines restructured counties and county-level cities as unified entities [11–13]. However, there are numerous variations in administration and finance between the pre-reformed counties and county-level cities [14–16]. Therefore, treating these two types of city-county mergers as homogeneous does not align with the principles of the difference-in-differences approach. From 1998 to 2007, given that the city-county merger implemented was mainly the TCD reform, we focus on the TCD reform and employ the DIDs method to examine the impact of TCD reform on firm-level TFP. In the robustness test, we also include counties transitioned into municipal districts in the treatment group.

Using county-level panel data from 1998–2007, we show that the TCD reform significantly decreases firm-level TFP. This finding survives an extensive series of robustness checks, including the parallel trend test, placebo test, substitution of the dependent variable, PSM-DID, CSDID, inclusion of regular counties in the regression, and exclusion of other policy interferences. Mechanism analysis indicates that the TCD reform predominantly influences firm-level TFP by modifying the firm's internal financial environment and the government's fiscal environment. The findings suggest that the city-county merger, which curtails the autonomy of reformed county-level government while bolstering the power of

prefecture governments, hurts the TFP of existing firms. This research sheds light on the complex interplay between government restructuring and firm-level productivity, and underscores the need for thoughtful consideration of the potential repercussions of such policy interventions on the economic and business landscape.

This study contributes to the existing body of literature on the implementation of amalgamation reforms at the sub-provincial administrative levels. Bo [17] explores the effects of amalgamation at the prefectural level on regional development through an analysis of China's reform of turning prefectures into prefecture-level cities in the 1980s. The findings of this research suggest that the transfer of political authority from prefectures to the newly established prefecture-level cities alleviates resource mismatch in the region and enhances industrial productivity at the city level. In addition, centralizing decision-making authority from counties to prefecture-level cities results in a clear core-periphery system, with prefecture-level city governments providing greater political and financial support to the counties within their jurisdiction, while those located on the periphery are often neglected [18]. This paper adds to the existing literature by highlighting the reconfiguration of power dynamics between prefecture-level cities and county-level cities through city-county mergers.

Furthermore, this paper broadens the research scope of municipal amalgamations. In recent decades, numerous countries have undergone municipal amalgamations. Several scholars have examined the correlation between municipal amalgamations and economic development, public services, and local finance [19]. Hall et al. [20] examine city-county mergers policy in the US and found that mergers do not always promote economic development and may even widen the gap between rural and urban areas. Matti and Neto [21] examine the economic impact of city-county mergers in the United States during the Great Depression and found that mergers did not enhance economic stability. Conversely, Blesse and Baskaran [4] find that municipal amalgamations can generate economies of scale using data from Germany. Reingewertz [3] reaches a similar conclusion in study of municipal amalgamations in Israel.

Additionally, some researchers have studied the impact of municipal amalgamation on government public services and finances. Hanes [22] and Roesel [23] find that the scale effect is an important influence on the effect of mergers. Besides, mergers can improve the efficiency of public expenditures and the quality of public services when the scale is below a critical threshold. Steiner and Kaiser [24] found that municipal amalgamations improve the quality of government public services, but the impact on government finances is not significant. However, Hinnerich [25], Rouse and Putterill [26] conclude that amalgamations do not improve the efficiency of government in providing public goods. Conversely, Hansen [27] finds that municipal amalgamations improve government finances.

In China, some research has suggested that the city-county merger could diminish interregional barriers and facilitate regional integration, thereby stimulating local economic growth [11]. However, Liu et al. [12] argue that the economic growth resulting from city-county mergers is only effective in the short term, lacking long-term impact, and could exacerbate regional inequalities. Apart from economic growth, studies have also explored the effects of city-county mergers on government finance, urban space utilization, housing prices and pollution emissions, respectively [10, 13, 28, 29]. Overall, previous research has mainly focused on the macro-level impacts of city-county mergers, with limited attention given to the effects on firm performance. Therefore, this paper contributes to extending the scope of research by looking at the impact of the city-county merger on firm-level productivity.

The rest of this paper is organized as follows: the section 2 provides the institutional background; section 3 provides the theoretical framework; the section 4 provides empirical strategy; the section 5 provides the empirical results and a series of robustness tests; the section 6 provides the mechanism analysis; and the section 7 is the conclusion.

## 2. China's city-county merger policy

After initiating the reform and opening-up policy, China implemented the "city-managed county" governance system, gradually establishing a four-tier administrative structure consisting of provinces, prefecture-level cities, counties, and townships (see Fig 1). County-level administrative units serve as a crucial link and represent the fundamental building block of China's economy [17]. Concurrently, the central government initiated a gradual transfer of economic management authority to local governments within the ambit of the "decentralization" framework. Consequently, the county not only functions as an independent level of governance but also serves as the fundamental administrative entity for decentralization reform. In this case, any alteration in county-level administrative structure would inevitably entail modifications in local administrative and financial powers, thereby influencing the local economy.

Starting in the 1980s, China developed a strategy for urbanization that focused on the growth of small and medium-sized cities. This led to the implementation of the county-to-city upgrading policy across the country, resulting in a significant increase in the number of county-level cities from 92 in 1978 to 442 in 1997. However, despite expanding the economic management, financial, and approval authorities of counties, the county-to-city upgrading reform did not yield the desired outcomes due to insufficient agglomeration of small cities [30, 31] and the absence of long-term incentives [15, 16, 32]. Consequently, in 1997, the Central Government decided to halt the county-to-city upgrading reform and shift the focus towards implementing city-county mergers. This policy involved transforming counties or county-level cities into municipal districts. Data published by the Ministry of Civil Affairs reveals that between 1998 and 2007, 112 city-county merger reforms took place in China's 28 provinces. Out of these, 38 reforms directly transformed counties into municipal districts, while 74 reforms turned county-level cities into municipal districts (as illustrated in Fig 2). Therefore, from 1998 to 2007, the transformation of county-level cities into municipal districts (TCD) emerged as the prevailing trend in the reform process.

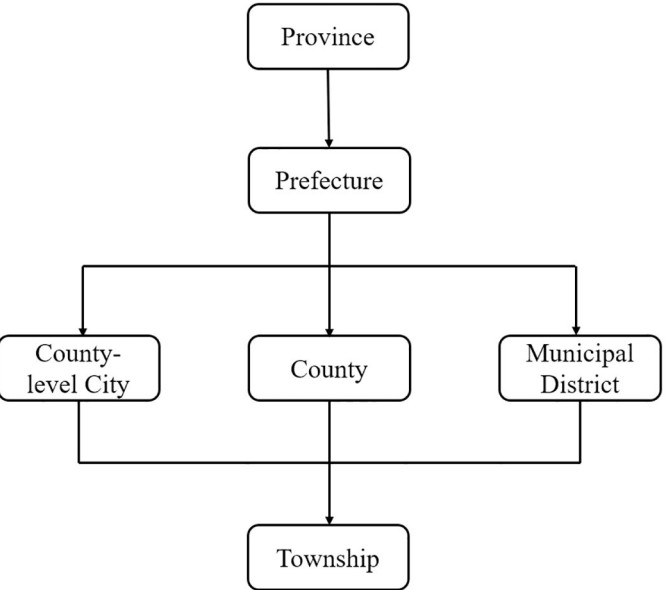

**Fig 1. China's local political hierarchies.**

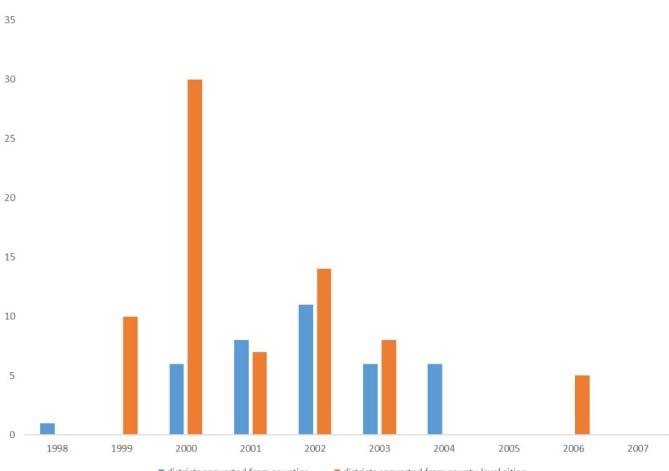

**Fig 2. Number of districts transferred from counties or county-level cities in 1998–2007.**

In 1998, there were 437 county-level cities, mainly located in the eastern part of China. By 2007, the number of counties had decreased to 368, a decrease of 69 over ten years. Moreover, in 2002, three new county-level cities were established. Therefore, the net reduction of county-level cities is 72, with 3 being revoked and 69 being transformed into municipal districts. Additionally, five county-level cities were transitioned into two separate municipal districts. Hence, the total number of municipal districts created from 1998 to 2007 is 74.

In contrast to the county-to-city upgrading reform, the city-county merger has a certain color of centralization. After being transitioned into municipal districts, the specific administrative and financial powers previously held by counties and county-level cities were centralized in the governments of prefecture-level cities, weakening the authority of the reformed counties [13]. Compared to counties and county-level cities, municipal districts are subservient to the prefecture-level cities. Put another way, municipal districts lack autonomy in infrastructure development, project approval, investment attraction, urban planning, industrial layout, etc., and are subject to the overall planning of the prefecture-level city [11]. Furthermore, regarding the financial system, the municipal districts have limited financial autonomy and are not autonomous, but instead depend on the prefecture-level city [14]. In this situation, the tax revenue of firms in the municipal districts is collected by the prefecture-level government before being allocated to the municipal districts. This leads to a decreased share of tax revenue for newly established municipal districts compared to pre-reformed counties and county-level cities. Moreover, when functioning as municipal districts, the allocation of fiscal expenditures should be adjusted to enhance people's welfare, with a decreased emphasis on infrastructure development.

## 3. Theoretical framework

When shifting from county or county-level cities to municipal districts, the authority of the county governments becomes partially centralized at higher levels of government. Despite municipal districts being in the same hierarchy as county or county-level cities, the transfers received by municipal districts are fewer than those received by pre-reformed county or county-level cities. Subsequently, the allocation of funds for infrastructure projects in these former county-level administrative units decreases when reformed as municipal districts

[11, 14]. This TCD reform weakens the local government's control over the "transaction efficiency".

The inframarginal model developed by Yang & Shi [33] is focused on enhancing "transaction efficiency", including the enhancements in both institutional transactional efficiency (e.g., decreasing corruption and safeguarding property rights) and non-institutional transactional efficiency (e.g., improvements in infrastructure development). Moreover, the inframarginal analytical framework is adept at illustrating the effects of changes in transaction efficiency on trade performance, production efficiency, and overall economic fluctuations. This model is capable of illuminating both the micro-level economic decisions made by firms (individuals) and macroeconomic operations.

As is explained in Section 2, the city-county merger involves shifts in local executive powers and subsequent changes in local infrastructure investment and construction. These adjustments notably influence transaction efficiency. Moreover, the impact of city-county mergers on productivity may be attributed to changes in both institutional and non-institutional transaction efficiencies. This study builds upon the inframarginal analytical framework developed by Yang & Shi [33] and expands the general equilibrium model to explore the theoretical implications of how the changes in transaction efficiency from county-city mergers affect firm-level total factor productivity (TFP).

Based on the inframarginal analysis of Yang & Shi [33], this paper develops an inframarginal model, presupposing that Eq (1) governs the optimization of consumption utility among typical consumers within a particular area:

$$U_{\max} = [X_1 + X_D + X_C]^{\frac{1}{\rho}} \tag{1}$$

Where $U$ represents the individual's utility level, $X_1$ is the set of self-sufficient goods produced by the individual, $X_D$ is the set of traded goods purchased by the individual, $X_C$ is the set of non-traded goods produced by the individual, and $\rho$ is the elasticity of substitution, $0 < \rho < 1$. Obviously, the inframarginal analysis includes self-sufficient products, traded goods, and non-traded goods simultaneously, implying that individuals can opt for either a corner or interior point solution. Especially, individuals can manifest as both producers and consumers, hereby encompassing both traditional consumer behavior and producer behavior within the neoclassical analytical framework.

Consumers' consumption pattern of self-sufficient products is set in the form of:

$$X_1 = x_1^\rho \tag{2}$$

Where $x_1$ is the self-sufficient product. And the consumption of traded goods by a typical consumer (firm) is defined as:

$$X_D = \sum_2^n (k x_d)^\rho \tag{3}$$

Where $x_d$ is the traded good, and $k$ represents transaction efficiency ($0 < k < 1$), encompassing institutional and non-institutional transaction efficiency. Specifically, a higher value of k denotes a more proficient transaction process, resulting in reduced transaction depletion and comparatively lower transaction expenses. Similarly, the consumption of non-traded goods is formulated as follows:

$$X_C = \sum_1^{m-n} x_c^\rho \tag{4}$$

Where $x_c$ is non-traded goods, $m$ denotes the number of differentiated varieties, $n$ is the number of traded goods consumed, and accordingly, $m - n$ is the consumption of non-traded varieties.

Individuals (firms)'s production process is subjected to the technological constrain in Eq (5):

$$x_1 + x_1^s = l_1^\alpha \tag{5}$$

where $l_1$ is the total inputs, $x_1^s$ denotes products manufactured by individual entities (firms) for commercial purposes, and $\alpha > 1$ indicates that individuals (firms) possess specialization advantages in the production of their respective offerings.

Technical constraint on autarkic goods is set as:

$$x_c = l_c \tag{6}$$

where $l_c$ is the labor input for non-traded goods.

Labor constraints and income constraints for individuals (firms) are defined in Eqs (7) and (8), respectively:

$$l_1 + \sum_1^{m-n} l_c = 1 \tag{7}$$

$$p_1^s x_1^s = \sum_2^n p_d x_d \tag{8}$$

The goods sold and purchased by a typical consumer are defined by $p_1^s$ and $p_d$, respectively. Eqs (7) and (8) imply that this economic system offers an optimal decision that integrates production and consumption, and can endogenously yield the optimal division of labor, productivity, variety of traded goods, etc. It is worth mentioning that these aspects remain unresolved within the neoclassical trade theory and heterogeneous firm trade theory.

According to Eqs (1)–(8), the relationship between productivity level $l_1^\alpha/l_1$ and transaction efficiency can be obtained as:

$$\left(1 - \frac{1}{1 + \frac{G(k)}{n-1}}\right) \frac{l_i^{\frac{\alpha\rho-1}{\rho-1}}}{1 - l_i} = (\text{m-n})^{-1} \alpha^{\frac{1}{1-\rho}} \tag{9}$$

Given that $\alpha\rho - 1 < 0$, since productivity $l_1^\alpha/l_1 = l_1^{\alpha-1}$ remains monotonic with $l_i$, the first term on the left side of the above equation reduces due to $G(k) = k^{\frac{\rho}{\rho-1}}$, where $dG(k)/dk < 0$. To maintain the equation unchanged, the second term on the left ($\frac{l_i^{\frac{\alpha\rho-1}{\rho-1}}}{1-l_i}$) needs to increase with $k$, implying that $dl_1/dk > 0$. In this case, $d\left(\frac{l_i^\alpha}{l_i}\right)\Big/dk > 0$.

Hence, this paper puts forward the following proposition:

*Transitioning a county or a county-level city into a municipal district involves partially centralizing power to the corresponding upper-level prefecture government. This power centralization process results in local governments shifting their focus away from enhancing transaction efficiency, which limits their ability to tailor policies and initiatives to the specific needs of the reginal economic development. Consequently, this shift can lead to a decline in firm-level productivity.*

## 4. Empirical strategy

### 4.1 Data

In this paper, TCD reform is constructed as a quasi-natural experiment to study the impact of China's administrative division adjustments on firm productivity. Therefore, this paper mainly draws from three data sources for empirical analysis. The first data source is managed by China's Ministry of Civil Affairs (MCA), which consists of county-level administrative division information. This data enables us to identify counties that have transitioned into municipal districts. The second one is the Annual Survey of Industrial Firms, providing us with firm-level total factor productivity and an array of firm-level and industry-level variables. The third source is the National Financial Statistics of Prefecture, Municipalities and Counties, from which we obtain macro-level data.

**4.1.1 China's Ministry of Civil Affairs (MCA).** MCA releases annual data on the adjustments made to administrative divisions in each province, enabling us to compile information on the TCD reform. In 1998, there were a total of 437 county-level cities, and out of these, 74 county-level cities underwent TCD reform between 1998 and 2007. Consequently, these 74 counties constitute the treatment group. Moreover, due to the repeal of Wanshang City in Yunnan Province in 1999, Qiongshan City in Hainan Province in 2002, and Miquan City in Xinjiang in 2007, these three county-level cities were ineligible for the DIDs method. Therefore, when summarizing the data, we excluded the samples from these three county-level cities.

**4.1.2 Annual Survey of Industrial Firms.** The Annual Survey of Industrial Firms, conducted by the National Bureau of Statistics of China, encompasses all state-owned manufacturing firms and firms of other ownership that have sales exceeding RMB 5 million (considered above-scale firms). From 1998 to 2007, the number of firms included in this survey increased from approximately 130000 to 310000. This database primarily contains firm-level production and financial data, which serves as a foundation for assessing TFP and other relevant control variables. For each firm, the survey records various information such as gross output value, sales figures, intermediate input, fixed assets, employment statistics, wage bills, registration year, capital structure, subsidies received, as well as details regarding the firm's location, ownership, and industry of operation. To ensure data quality, we apply the following criteria for data cleaning: (1) exclusion of firms with missing data on firm identification code, firm name, employment, intermediate input, value-added, output, and fixed assets; (2) elimination of firms with fewer than nine employees; (3) exclusion of firms where the intermediate input or value-added exceeds the output; (4) removal of firms with negative values for intermediate inputs, value added, output, and fixed assets. After data cleaning, we obtained a total of 1,939,520 observations, comprising 663,051 firms.

**4.1.3 National Financial Statistics of Prefecture, Municipalities and Counties.** This database contains detailed information on fiscal revenues, expenditures, and major economic indicators for all levels of administrative units in China, making it an ideal data source for studying the impact of TCD policy on local government finances. Specifically, the county-level data used in this paper, including GDP, population, gross expenditure, capital expenditure, public security expenditure, specific subsidies, and revenue from government-managed funds, is derived from this database.

**4.1.4 Merged data.** We combined the panel data of city-level counties and the Annual Survey of Industrial Firms by the six-digit administrative division code recorded in both databases from 1998 to 2007. The merged database contains a total of 524,827 observations, with 189,562 firms included. The county-level cities in this sample collectively contribute to 22.9% of the total industrial output value, while their sales revenue accounts for 23.5% of the sales of the total sample.

## 4.2 Empirical specification

Given that the reforms in different regions took place at different times, we employ the multi-period DIDs approach to examine the effects of TCD reform on firm-level Total Factor Productivity (TFP). The fundamental regression model is represented by Eq (10):

$$TFP_{ijt} = \beta_0 + \beta_1 policy_{it} + \beta_2 X_{ijt} + \delta_i + \eta_j + \mu_t + \varepsilon_{ijt} \tag{10}$$

Where $TFP_{ijt}$ is the TFP of firm $i$ in industry $j$ in year $t$. $policy_{it}$ indicates the policy variable of TCD reform, which is the combined effect of the treatment indicator ($treat_i$) responding to the TCD reform and the post-reform indicator ($post_t$) representing the post-TCD reform period. When the reform takes place in the county-level city, $treat_i$ equals 1 (treatment group), otherwise it is 0 (control group); when the TCD reform occurs in the county-level city in year $i$ and the subsequent years, $post_t$ takes the value of 1, otherwise it is 0. $X_{ijt}$ consists of a set of control variables at the firm and industry level; $\delta_i$, $\eta_j$ and $\mu_t$ denote firm, industry, and time fixed effects, respectively; $\varepsilon_{ijt}$ is the error term.

The variable of interest is the firm-level total factor productivity (TFP), which brings about estimating challenges due to endogeneity issues generated by the conventional ordinary least squares (OLS) method. As a result, previous studies have primarily relied on the O-P [34] and L-P [35] approaches to estimate TFP. To align with the existing research, we also approach the L-P and O-P estimations for measuring firm-level TFP [36]. To avoid the omission of observations with zero investment when estimating TFP by the O-P method, which could considerably reduce the sample size, we employ the L-P approach to estimate firm-level TFP in the baseline regression. Furthermore, we also use the O-P method to estimate the firm-level TFP as an alternative dependent variable in the robustness tests.

This paper incorporates a range of control variables, including firm age, the square term of firm age, export status, foreign ownership status, firm size as indicated by main business revenue, industry concentration measured by the HHI, the square of HHI, and the logarithm of subsidies received by the firm, as well as the square of subsidies. Descriptive statistics of the control variables are shown in Table 1:

## 5. Results

### 5.1 Basic results

The results obtained from specification (10) are presented in Table 2. We report the findings in column (1) without incorporating control variables and industrial fixed effect. The coefficient of $policy_{it}$ is significantly negative at -0.038. When firm-level control variables are taken into account, the coefficient slightly decreases to -0.041, as indicated in column (2). In column (3), the industrial fixed effect is included, resulting in a significantly negative coefficient of -0.038. Our preferred specification, including firm- and industry-level control variables, is displayed in column (4). The coefficient is significantly negative at 0.042, suggesting that, on average, there is a 4.2% decrease on firm-level TFP in the reformed regions compared to those regions that have not undergone TCD reform. Hence, the basic findings demonstrate that the TCD reform leads to a decline in the TFP of local firms, which aligns with the theoretical model of this paper. The results indicate that the reform causes a decrease on firm-level TFP due to reduced "transaction efficiency".

### 5.2 Robustness tests

In the baseline regression, this study confirms that the TCD reform will result in a reduction on firm-level TFP. However, it is important to consider other potential variables that could

**Table 1. Descriptive statistics of the relevant variables.**

| variables | Definition | N | mean | sd |
|---|---|---|---|---|
| | Firm level variables | | | |
| TFP | Firm's TFP calculated by LP method | 488,916 | 5.605 | 1.056 |
| TFP_OP | Firm's TFP calculated by OP method | 488,916 | 3.103 | 0.983 |
| age | Logarithm of firm's age | 507,008 | 1.819 | 0.931 |
| age2 | Logarithm squared of firm's age | 507,008 | 4.174 | 3.586 |
| ex | Whether firm export or not | 524,827 | 0.288 | 0.453 |
| soe | SOEs or not | 524,827 | 0.0619 | 0.241 |
| foreign | foreign-funded firm or not | 524,827 | 0.131 | 0.337 |
| lnsub | Logarithm of firm's subsidies | 524,562 | 1.197 | 1.578 |
| lnsub2 | Logarithm squared of firm's subsidies | 524,562 | 3.925 | 12.19 |
| scale | Firm's scale | 524,827 | 0.624 | 3.876 |
| labor | Logarithm of firm's employees | 524,827 | 4.744 | 1.060 |
| tax | Logarithm of firm's tax | 524,355 | 2.934 | 2.834 |
| addvalue | Logarithm of firm's add value | 519,316 | 8.635 | 1.319 |
| rota | Return on total assets | 524,653 | 0.0964 | 0.276 |
| fc | Constraints on enterprise financing | 490,465 | -2.489 | 1.466 |
| | Regional level variables | | | |
| policy | TCD reform | 524,827 | 0.184 | 0.387 |
| policy1 | Province-managing-counties | 524,827 | 0.0514 | 0.221 |
| policy2 | Expanding power to strengthen counties | 524,827 | 0.290 | 0.454 |
| nyr | Agriculture as a share of GDP | 3,331 | 0.230 | 0.163 |
| gyr | Industry as a share of GDP | 3,331 | 0.475 | 0.176 |
| ppgdp | Per capita GDP | 3,172 | 1.369 | 1.339 |
| rsoe | The proportion of SOEs' add value in the province | 3,331 | 0.00330 | 0.0100 |
| rfor | The proportion of foreign-funded add value in the province | 3,331 | 0.00159 | 0.00528 |
| dage | Firms' age in the region | 3,331 | 14.24 | 55.84 |
| dage2 | Logarithm of firm's age in the region | 3,331 | 3,320 | 98,461 |
| spending | Logarithm of Gross expenditure per capita | 3,171 | 6.552 | 0.707 |
| capital | Logarithm of capital expenditure per capita | 1,951 | 2.863 | 1.459 |
| public | Logarithm of public security expenditure per capita | 2,811 | 3.678 | 0.726 |
| subsidies | Logarithm of specific subsidies per capita | 2,805 | 4.284 | 0.842 |
| fund | Logarithm of per capita revenue from government-managed funds | 3,171 | 6.695 | 0.731 |
| | Industry level variables | | | |
| hhi | HHI | 524,827 | 0.0165 | 0.0295 |
| hhi2 | Squared of HHI | 524,827 | 0.00114 | 0.00987 |

affect the regression outcomes. Hence, in order to ensure the reliability of the research findings, this paper carries out a set of robustness checks.

**5.2.1 Flexible estimation.** To demonstrate the validity of the DIDs approach, we employ a flexible estimation technique to evaluate the common trend between the treated and control groups before the TCD reform. A crucial assumption when utilizing DIDs to assess the impact of a policy is to satisfy the hypothesis, which posits that the treated and control groups are expected to exhibit similar patterns of change before the policy intervention, while displaying divergent trends afterward. The time frame examined in this study spans from 1998 to 2007, encompassing up to eight periods of pre- and post-reform observations as all reforms occurred post-1998 and no reforms took place in 2007. On this basis, the flexible estimation formulated

**Table 2. Baseline regression results.**

| | (1) | (2) | (3) | (4) |
|---|---|---|---|---|
| | TFP | TFP | TFP | TFP |
| policy | -0.040*** | -0.041*** | -0.040*** | -0.042*** |
| | (-3.11) | (-3.25) | (-3.17) | (-3.31) |
| age | | 0.184*** | | 0.183*** |
| | | (24.31) | | (24.35) |
| age2 | | -0.043*** | | -0.043*** |
| | | (-18.66) | | (-18.75) |
| ex | | 0.089*** | | 0.089*** |
| | | (16.92) | | (16.98) |
| scale | | 0.021*** | | 0.021*** |
| | | (2.97) | | (2.96) |
| foreign | | 0.010 | | 0.010 |
| | | (1.08) | | (1.08) |
| hhi | | -0.643*** | | -1.125*** |
| | | (-4.43) | | (-6.14) |
| hhi2 | | 1.449*** | | 2.237*** |
| | | (3.81) | | (4.37) |
| lnsub | | -0.021*** | | -0.021*** |
| | | (-5.39) | | (-5.48) |
| lnsub2 | | 0.005*** | | 0.005*** |
| | | (9.11) | | (9.18) |
| Constant | 5.671*** | 5.483*** | 5.671*** | 5.490*** |
| | (2,503.91) | (560.95) | (2,512.50) | (557.50) |
| Firm FE | YES | YES | YES | YES |
| Year FE | YES | YES | YES | YES |
| Industry FE | NO | NO | YES | YES |
| Observations | 411,575 | 411,575 | 411,575 | 411,575 |
| R-squared | 0.788 | 0.790 | 0.788 | 0.791 |

Note:

*, **, and *** indicate significant at 10%, 5%, and 1% significance levels, respectively, with t-values in parentheses, below.

in this paper is described below:

$$TFP_{ijt} = \beta_0 + \sum_{p \geq -S, \, p \neq -8}^{L} \beta_p D_{ijt}^p + \beta_2 X_{ijt} + \delta_i + \eta_j + \varepsilon_t + \mu_{ijt} \qquad (11)$$

Let $T_i$ represent the year when the county where firm $i$ is located initiates TCD reform, where S and L are the maximum number of years tested before and after $T_i$. We define $D_{ijt}^p = 1$ if $t - T_i = p$, and 0 otherwise, for $-S \leq p \leq L$ and $p \neq -8$. To illustrate, we used the initial year of the sample period as the base and computed the coefficients for all other years relative to this base. Fig 3 shows the coefficients and their 95% confidence interval of Eq 11. If the coefficient is negative (positive) in Fig 4, it signifies a decrease (increase) in TFP for the treated group compared to the control group within a particular year. If the 95% confidence interval encompasses zero, the coefficient lacks statistical significance (at a 5% significance level). It can be observed that prior to the implementation of the TCD reform, the coefficients of the treated and control groups exhibited insignificance, indicating no substantial disparity on the

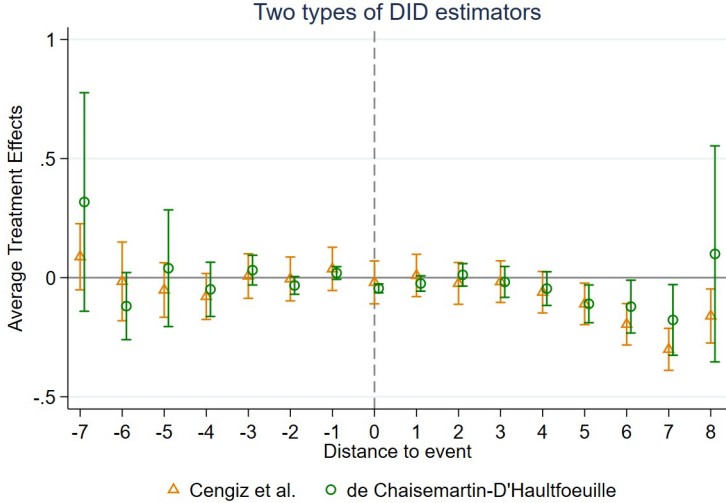

**Fig 3. Heterogeneous treatment effects.**

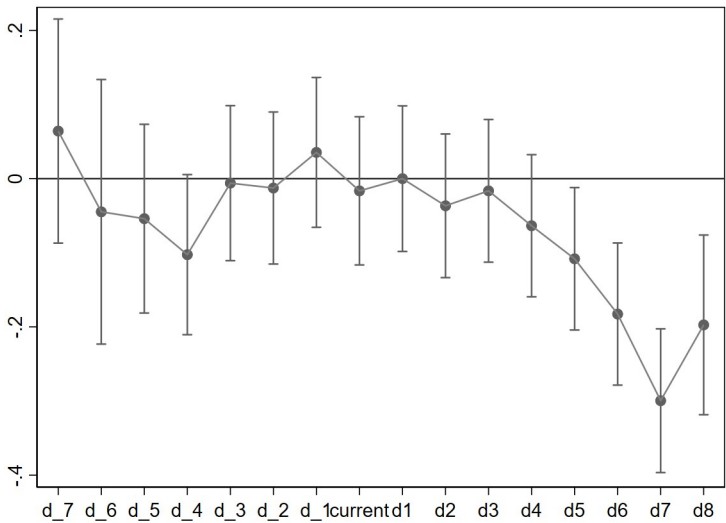

**Fig 4. Parallel trend test.**

TFP of firms in the reformed and unreformed county-level cities. Hence, the assumption of a common trend holds.

Fig 4 also demonstrates that the influence of TCD reform on firm-level TFP does not occur immediately following the policy. In the initial four years following the reform, the treated group experiences a decline in TFP compared to the control group. Nevertheless, this decrease is not statistically significant at a 5% level. The major impact, however, materialize after the lapse of four years. This could be attributed to the relatively lengthy infrastructure construction cycle, resulting in the continued development of projects that are initiated before the reform. Consequently, the significantly negative effects on firm-level TFP have been postponed. However, with the passage of time, the diminished authority of the county government following the reform has impeded transactional efficiency, resulting in a significant decrease on firm-

level TFP. Additionally, on average, the impact of TCD on firm-level TFP diminishes from the fifth year to the final year in our sample. These findings suggest that our results may not fully capture the long-term effects and consequently underestimate the influence of TCD reform.

**5.2.2 Placebo testing.** To eliminate the influence of the TCD reform on firm-level TFP originates from undisclosed factors, this study conducts a placebo test. Following the approach of Li et al. [37], policy pilots and time of policy shock should be randomly chosen and combined into cross-multipliers for the analysis. To illustrate, we randomly select 74 county-level cities out of the 437 county-level cities in 1998 and select the time of TCD reform from 1999 to 2006 for 500 times. For each time, the difference-in-differences specification, as defined in Eq (10), is estimated for the whole sample. In this case, we derive 500 estimated coefficients for $policy_{it}$. The rationale behind this placebo test is that the 500 estimated coefficients for $policy_{it}$ should show no significant difference from zero, as they do not represent the real reforms. As depicted in Fig 5, the results of the placebo test indicate that the coefficients of $policy_{it}$ in the 500 regressions predominantly cluster around zero, forming a roughly normal distribution in the density function. Hence, the placebo test provides evidence that our estimated impact of the TCD reform on firm-level TFP, obtained through the basic regression, is unlikely to be influenced by other unobservable factors.

**5.2.3 Alternative independent variable.** Difference-in-Differences settings rely on the crucial assumption that the chosen sample should possess homogeneity in all aspects except for the impact of the policy shock. Hence, even though there were instances of counties and county-level cities transitioning to municipal districts during the sample period, our focus remains solely on evaluating the effects of TCD reform on county-level cities to uphold homogeneity in our basic regression. Nevertheless, it is worth noting that a total of 38 counties in China underwent conversion into districts, alongside county-level cities, between 1998 and 2007. To thoroughly examine the scenario throughout the entire sample, we also conducted additional tests considering both counties and county-level cities that underwent this transition as the treated group.

The corresponding findings presented in Table 3 reveal that the coefficient of $policy_{it}$ remains statistically significant in the full-sample test. Our preferred specification in column (4) indicates that the coefficient of $policy_{it}$ remains negative at 1% significance level. Therefore,

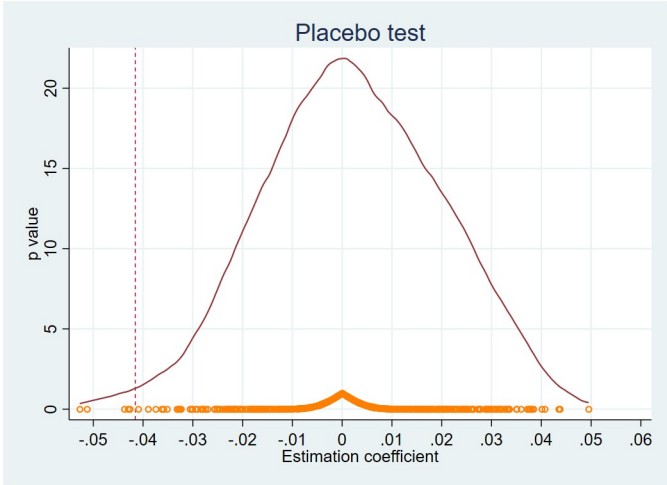

**Fig 5. Placebo testing.**

**Table 3. Full sample test.**

|  | (1) | (2) | (3) | (4) |
|---|---|---|---|---|
|  | TFP | TFP | TFP | TFP |
| policy | -0.026** | -0.030*** | -0.027** | -0.031*** |
|  | (-2.35) | (-2.75) | (-2.45) | (-2.86) |
| Constant | 5.656*** | 5.105*** | 5.656*** | 5.111*** |
|  | (2,315.32) | (422.83) | (2,325.93) | (421.79) |
| Control variables | NO | YES | NO | YES |
| Firm FE | YES | YES | YES | YES |
| Year FE | YES | YES | YES | YES |
| Industry FE | NO | NO | YES | YES |
| Observations | 459,279 | 459,279 | 459,279 | 459,279 |
| R-squared | 0.776 | 0.782 | 0.777 | 0.783 |

the policy of TCD reform continues to have a notable adverse influence on firm-level TFP when adding all the reformed counties and county-level cities into the sample.

**5.2.4 PSM-DID.** When implementing the TCD reform policy, the selection of reformed county-level cities is not purely random but instead based on specific characteristics [12]. Therefore, it is crucial to consider the selective bias that may arise from this sample selection and its impact on the study results. Therefore, this paper employs the PSM-DIDs method for robustness checks. Currently, there are two primary approaches to applying the PSM-DIDs model in panel data: one treats the entire panel data as cross-sectional data, while the other matches each period in the panel data on a yearly basis. Both methods are widely used but possess their own advantages and disadvantages. As a result, we opt for employing both approaches to ensure matching accuracy and robustness. Specifically, the covariates selected include the total population of the region, the percentage of non-agricultural population, GDP per capita, the proportion of industrial output to total output value, and per capita financial income. Subsequently, we apply nearest-neighbor matching based on the two aforementioned methods. Only the matched samples that fulfill the criteria of common support are subjected to multi-period difference-in-differences. The relevant results are presented in Table 4, where columns (1)-(2) show the PSM-DIDs regression results matched according to the cross-sectional method, and columns (3)-(4) show the PSM-DIDs results matched on a year-by-year

**Table 4. PSM-DID test.**

|  | (1) | (2) | (3) | (4) |
|---|---|---|---|---|
|  | TFP | TFP | TFP | TFP |
| policy | -0.032** | -0.036*** | -0.049*** | -0.058*** |
|  | (-2.47) | (-2.80) | (-2.66) | (-3.13) |
| Constant | 5.670*** | 5.489*** | 5.686*** | 5.496*** |
|  | (3,637.61) | (528.70) | (2,651.11) | (496.71) |
| Control variables | NO | YES | NO | YES |
| Firm FE | YES | YES | YES | YES |
| Year FE | YES | YES | YES | YES |
| Industry FE | YES | YES | YES | YES |
| Observations | 375,918 | 375,918 | 280,467 | 280,467 |
| R-squared | 0.791 | 0.794 | 0.796 | 0.800 |

basis. Regardless of the methods employed, it is evident that the coefficient of the core explanatory variable $policy_{it}$ is significantly negative.

**5.2.5 Heterogeneous treatment effects.** Since the TCD reform was instituted at varying time points across different regions, this study employs the multi-period DIDs method in the baseline regression. However, the multi-period DIDs approach encounters heterogeneous treatment effects, which could introduce bias and inconsistency in the estimates derived from traditional two-way fixed effects models (TWFE) [38, 39]. On the one hand, post-processed samples might mistakenly treat pre-processed samples as a control group [39]. On the other hand, the estimated coefficients for each period may also interact with one another [40]. To address the potential bias of the TWFE estimator in the multi-period DIDs analysis, this paper incorporates the CSDIDs method proposed by Callaway & Sant'Anna [41] for robustness checks. Initially, two approaches were employed to select the control group. One approach involves selecting samples that have never undergone treatment as the control group. The other approach involves selecting samples that have never been treated and have not yet received any treatment as the control group. Subsequently, the average treatment effect (ATT) was computed for each group period. Finally, the average treatment effects for each group-period were aggregated to obtain an estimate of the overall average treatment effect. The outcomes of the robustness checks are presented in Table 5, where ATT1 captures the average treatment effect when treating the never-treated samples as a control group, and ATT2 represents the average treatment effect when treating the never-treated and not yet treated samples as a control group. It is evident that both methods yield significantly negative average treatment effects, thereby affirming the robustness of the baseline regression results.

To further demonstrate the robustness of the conclusions in this paper, we incorporate the DIDM estimator proposed by De Chaisemartin & D'Haultfoeuille [38] as well as the Stacked Regression Estimator proposed by Cengiz at al. [42] to rectify the Staggered DID and exhibit dynamic treatment effects. The test results are depicted in the figure as follows, indicating no significant difference between the coefficients of the treatment and control groups in periods prior to the policy, further bolstering the robustness of the baseline regression results.

**5.2.6 Exclusion of other factors.** In the main analysis, we use firm-level TFP estimated from the methodology proposed by Levinsohn and Petrin [35]. This approach uses specific materials to control for unobserved productivity. To further validate our findings, we also estimate firms' TFP employing the approach developed by Olley and Pakes [34]. This alternative methodology incorporates investment to control unobserved productivity and considers firm exit. The outcome is displayed in column (1) of Table 6. It becomes evident that when we modify the measurement method of the explanatory variables, the estimated coefficient of the main explanatory variable, policy, remains significantly negative at the 1% significance level. This observation indicates that our basic results remain unchanged regardless of the measurement method utilized for TFP.

In the process of restructuring urban administrative areas, numerous counties have acquired the county-level city status, while others have been transitioned into city districts. Regardless of the transaction methods, the reformed counties are still administered by the corresponding prefecture cities. Therefore, these modifications have undoubtedly provided

**Table 5. CSDID test.**

|  | (1) | (2) | (3) | (4) | (5) |
|---|---|---|---|---|---|
|  | Coefficient | sd | Z-value | P-value | Observations |
| ATT1 | -0.0375** | 0.017457 | -2.15 | 0.032 | 301085 |
| ATT2 | -0.0382** | 0.017453 | -2.19 | 0.029 | 301166 |

**Table 6. Exclusion of other factors.**

| | (1) | (2) | (3) | (4) | (5) | (6) | (7) | (8) |
|---|---|---|---|---|---|---|---|---|
| | TFP OP | TFP | TFP | TFP | TFP | TFP | TFP | TFP |
| policy | -0.034*** | -0.043*** | -0.028** | -0.029** | -0.039*** | -0.026* | -0.040*** | -0.042* |
| | (-2.76) | (-3.40) | (-2.17) | (-2.26) | (-3.10) | (-1.91) | (-3.19) | (-1.71) |
| policy1 | | -0.043*** | | -0.041*** | | | | |
| | | (-6.46) | | (-6.31) | | | | |
| policy2 | | | 0.080*** | 0.080*** | | | | |
| | | | (11.36) | (11.30) | | | | |
| Constant | 3.067*** | 5.493*** | 5.465*** | 5.468*** | 5.469*** | 5.483*** | 5.490*** | 5.490*** |
| | (328.44) | (557.54) | (549.67) | (549.82) | (552.29) | (551.57) | (557.40) | (281.76) |
| Control variables | YES | YES | YES | YES | YES | YES | YES | YES |
| Year*2-digit industry code FE | NO | NO | NO | NO | YES | NO | NO | NO |
| County-level city*2-digit industry code FE | NO | NO | NO | NO | NO | YES | NO | NO |
| province*2-digit industry code FE | NO | NO | NO | NO | NO | NO | YES | NO |
| Firm FE | YES | YES | YES | YES | YES | YES | YES | YES |
| Year FE | YES | YES | YES | YES | YES | YES | YES | YES |
| Industry FE | YES | YES | YES | YES | YES | YES | YES | YES |
| Observations | 411,575 | 411,575 | 411,575 | 411,575 | 411,575 | 411,575 | 411,575 | 411,575 |
| R-squared | 0.752 | 0.791 | 0.791 | 0.791 | 0.793 | 0.795 | 0.792 | 0.791 |

benefits to the corresponding central cities. To overcome the limitations of the "city-managing-county" system, China also implemented two decentralization reforms within the given timeframe: the administrative reform known as "strengthen counties by expanding their powers" (*Kuo quan qiang xian*) and the fiscal reform known as "province-managing-county" (*Sheng zhi guan xian*). The goal of "strengthen counties by expanding their powers" is to divert some authorities from the central city to the county-level cities while maintaining the existing administrative structures. This approach aims to reduce administrative levels, enhance work efficiency, and lower administrative costs. "Province-managing-county" broadens the fiscal independence of counties, eradicating the prefecture city administration's role as the intermediary level connecting the province and the county [43].

Although these two reforms do not directly impact on the treatment group, they influence the control group as they entail modifications in the administrative and financial authority of counties. Thus, this paper includes the above two reforms as control variables. Columns (2)-(4) of Table 6 showcase the impacts when incorporating the "province-managing-county" reform (*policy*1), the "strengthen counties by expanding their powers" reform (*policy*2), and both reforms to the basic regression. The findings reveal that even after including both reforms, the negative influence of the TCD reform on firm-level TFP remains significant, supporting the robustness of our basic results. The estimated coefficients for *policy*2 are positive in both regressions, indicating that empowering county governments, and expanding their economic and social management authority have a positive impact on enhancing local firm-level TFP.

It's worth noting that "province-managing-county" reform causes a decrease on firm-level TFP, even though it is a decentralized reform. The possible reason is that policy1 centers on the allocations of fiscal revenues and expenditures to establish a direct connection between the province and the county, thereby increasing the fiscal authority of county governments and eliminating the role of prefecture city administration as an intermediary between the province and the county. More specifically, policy1 can be seen as a type of governmental management

restructuring that shifts from a "province-city-county" management system to a "province-county" management system. This restructuring amplifies the fiscal revenue of counties. However, given that the number of counties far exceeds the number of prefecture-level cities, policy1 greatly increases the number of administrative units that the province needs to manage horizontally. This escalation in the level of management span diminishes the supervisory and coordinating capabilities of the provincial government, leading to a rise in illegal activities such as land corruption and misappropriation of public funds [17, 37]. Therefore, although policy1 expands the financial power of counties and improves their financial income, it lowers the efficiency of county governments, consequently lowering firm-level TFP.

Moreover, in the preceding section, the placebo test partly mitigates the impact of unidentified factors on the findings. However, there may still be other policies that could impact the results. As most policies are implemented on a regional or industrial basis, we further incorporate interactions between regions and industrial fixed effects, as well as interactions between year dummies and industrial fixed effects, in columns (5)-(7) of Table 6. The results exhibit robustness.

In the baseline regressions, this paper uses robust standard errors clustered to the firm level. To mitigate the potential impact of clustering issues on the regression results, this paper conducts robust standard errors at county level and apply the Wild Cluster Bootstrap [44] for further robustness checks. In column (8) of Table 6, the coefficient of variable of interest (policy) is statistically significant at 10% level after clustering to the county level. Moreover, the implementation of the Wild Cluster Bootstrap method further supports the robustness of our benchmark results, with the coefficient of policy showing significance at 10% level.

**5.2.7 Heterogeneity analysis.**   China is a vast nation that displays considerable variations between regions. Throughout the last forty years of China's policies promoting openness, the coastal regions have attracted international enterprises owing to their convenient accessibility, affordable workforce, political advantages, and promising market prospects. In contrast, the central and western regions lag behind the eastern region in terms of infrastructure, economic connections, population density, capital availability, and technological advancements. Therefore, the eastern counties enjoy a stronger economic position. Although inland regions have also opened up to foreign businesses in recent years, they still do not possess a competitive advantage over their coastal counterparts. As a result, there exists a significant disparity in economic development between China's eastern region and the central and western regions. Consequently, it is reasonable to expect regional variations in the impact of the TCD reform on firm-level TFP.

We carry out a regional heterogeneity test by dividing the samples into two groups—firms located in the eastern region and firms located in the central-western region—to check the regional heterogeneity. The findings presented in columns (1)-(2) of Table 7 illustrate the impact of the TCD reform policy on firm-level TFP in the eastern and central-western regions, respectively. The analysis reveals that the influence of the TCD reform on firm-level TFP in the eastern region lacks statistical significance. Conversely, in the central-western regions, the impact on firm-level TFP is significantly negative with a significance level of 1%. It is logical to argue that the TCD reform has a limited impact on the firm-level TFP in the eastern regions, as these regions already possess well-established infrastructure. Consequently, the consolidation of power does not significantly affect transaction efficiency in the reformed county-level cities. However, in the central-western regions, where the city administration is responsible for the comprehensive growth of the city, there is a chance that the specific requirements of individual districts might be disregarded. This oversight can result in a decline in the ability of reformed county-level cities to improve transaction efficiency as they transfer some of their authority to the municipal government.

**Table 7. Heterogeneity analysis.**

|  | eastern | central-western | SOEs | Non-SOEs |  |  |
|---|---|---|---|---|---|---|
|  | (1) | (2) | (3) | (4) | (5) | (6) |
|  | TFP | TFP | TFP | TFP | TFP | TFP |
| policy | -0.020 | -0.155*** | -0.005 | -0.049*** | -0.045*** | -0.022* |
|  | (-1.40) | (-5.19) | (-0.11) | (-3.78) | (-3.59) | (-1.86) |
| policy*size |  |  |  |  | 0.014*** |  |
|  |  |  |  |  | (2.66) |  |
| policy*age |  |  |  |  |  | -0.065*** |
|  |  |  |  |  |  | (-8.64) |
| Constant | 5.516*** | 5.343*** | 4.819*** | 5.541*** | 5.510*** | 5.482*** |
|  | (525.31) | (202.46) | (61.40) | (552.60) | (604.05) | (573.42) |
| Control variables | YES | YES | YES | YES | YES | YES |
| Firm FE | YES | YES | YES | YES | YES | YES |
| Year FE | YES | YES | YES | YES | YES | YES |
| Industry FE | YES | YES | YES | YES | YES | YES |
| Observations | 343,720 | 67,848 | 21,991 | 385,458 | 411,575 | 411,575 |
| R-squared | 0.787 | 0.805 | 0.852 | 0.777 | 0.790 | 0.793 |

Mo, this paper also examines the influence of the TCD reform on firm-level TFP based on their ownership structure. State-owned enterprises (SOEs) have closer political and economic ties with local governments [45, 46], which means that they can benefit from more supportive policies, including special government subsidies, preferential treatment in government projects, access to credit facilities, etc. However, non-SOEs receive limited resources from the government compared to SOEs, apart from general policy support. In this scenario, firms with varying ownership structures may respond differently to policy shocks. Therefore, we classify firms into SOEs and non-SOEs to examine the impact of the TCD reform on firm-level TFP across different ownership types. The corresponding results are outlined in columns (3)-(4) of Table 7. Column (3) indicates that the policy has insignificant impacts on SOEs, potentially due to the policy favoritism that protects them against the adverse consequences of reduced transaction efficiency resulting from the TCD reform. However, in column (4), the TFP of non-SOEs was considerably diminished as a result of the TCD reform due to the lack of political affiliations.

In addition, this paper explores the impact of the TCD reforms on firms while considering the heterogeneity of firm sizes. To illustrate, comparing to smaller firms, larger firms possess greater assets, higher revenue, more investment in innovation, and closer connections to the government. To test whether the impact of TCD reforms on firm-level TFP differs based on firm sizes, we construct the cross-multiplier term of policy and firm size (*policy*size*). The relevant findings are shown in column (5) of Table 7. The coefficient of *policy*size* is significantly negative, indicating that an increase in firm size mitigates the impact of the TCD reforms on firm-level TFP, that is, smaller firms are impacted more by the reform.

Several studies have indicated out that firm performance is associated with firm age [47, 48], and firms of different ages may respond differently to adjustments in the external environment [49]. Therefore, this paper further examines whether the TFP reduction effect of TCD reforms varies with the age of firms. this paper constructs the cross-multiplier term between policy and firm age (*policy*age*) to measure whether the impact of TCD reforms on firm-level TFP differs based on firm ages. The relevant results are shown in columns (6) of Table 7. The correlation between *policy*age* and TFP is significantly negative, indicating that older firms

are more negatively affected by reforms. This could be attributed to the fact that older firms have established stronger connections with the government over time, making them more politically connected and susceptible to policy preferences compared to younger firms. Consequently, reforms aimed at reducing county government influence and support for companies have a more pronounced effect on older firms, resulting in a greater decline in TFP. Note that this result does not contradict our previous findings that TCD reforms insignificantly affect SOEs with stronger political connections. Due to their special status in China, SOEs will always receive priority support from local governments, regardless of TCD reforms.

## 6. Mechanism analysis

Previous analyses mainly examine the adverse consequences of transitioning county-level cities to city districts on firm-level total factor productivity, and the conclusion remains robust across different scenarios. However, the TCD reform is a policy aimed at modifying the administrative divisions. Hence, it is crucial for us to identify the mechanism through which the TCD reform can impact firm-level TFP. In section 3 of our study, our theoretical model suggests that this reform often leads to an upward shift of power to the corresponding municipal city, while diminishing the administrative and financial authority of the reformed county-level cities. As a result, there is a subsequent decrease in government revenue, expenditure, and the provision of public goods in reformed county-level cities, which ultimately affects the "transaction efficiency". Thus, the TCD reform has a profound impact on the external environment for firms operating in these reformed counties. Accordingly, it is likely to affect firms' internal financial conditions as well. Therefore, our analysis will focus on the changes in government revenue and expenditure, as well as modifications in firms' financial positions following the policy shock, as the mechanism by which the TCD reform impacts firm-level TFP.

### 6.1 Impact of the TCD reform on the government's financial situation

Firms always adapt their production strategy to the regional policy shocks. The fluctuations in local fiscal balance following the TCD reform are likely to affect the external circumstances for firms. Thus, a concern is that the alterations in county-level cities' fiscal balances after shock may serve as the conduit through which the TCD reform influences firm-level TFP. Table 8 provides an illustration of the impact of the TCD reform on the government's fiscal balance.

Columns (1)-(3) of Table 8 indicate that there has been a decrease in total expenditure per person (spending), per capita expenditure on infrastructure development (capital), and per capita expenditure on public prosecution and law enforcement (public) within government spending. Notably, the most significant decline was observed in per capita expenditure on infrastructure development (capital), which witnessed a substantial reduction of 35.6 percent.

This indicates that the financial autonomy of the county governments declines significantly after reform [11]. In addition, there is a reduction in the government's productive spending. Furthermore, the authority over infrastructure development, initially held by the county governments, is transferred to the central city. However, the city government shows insufficient attention towards the newly established municipal districts located on the outskirts of urban areas [18]. Consequently, the investment on infrastructure construction experienced a greater decrease compared to the pre-reform era. Thus, the reduction in provision of some essential public services, which were formerly delivered by local authorities, has had a holistic effect on the "transaction efficiency" within the region, thereby impacting the firm-level TFP. Moreover, considering government revenues, columns (4)-(5) reveal a 12.4% decrease in per capita government fund revenues (fund) and a 17.3% decline in per capita special subsidies (subsidies). This signifies that the reformed county-level cities experienced a reduction in government

**Table 8. Changes in the government's financial situation.**

|  | (1) | (2) | (3) | (4) | (5) |
|---|---|---|---|---|---|
|  | spending | Capital | public | fund | subsidies |
| policy | -0.100*** | -0.356** | -0.269*** | -0.124*** | -0.173*** |
|  | (-3.17) | (-2.31) | (-4.47) | (-3.05) | (-2.63) |
| nyr | -0.144** | -0.205 | 0.085 | -0.199*** | 0.167 |
|  | (-2.18) | (-0.55) | (0.97) | (-2.67) | (1.24) |
| gyr | 0.060 | 0.336 | 0.015 | 0.068 | -0.142* |
|  | (1.33) | (1.43) | (0.28) | (1.39) | (-1.79) |
| ppgdp | 0.054*** | -0.029 | 0.066*** | 0.048*** | 0.076*** |
|  | (5.04) | (-0.45) | (4.17) | (4.95) | (3.09) |
| rsoe | 1.964** | 8.811** | 1.446 | 2.203* | 0.011 |
|  | (2.10) | (2.08) | (1.39) | (1.88) | (0.01) |
| rfor | 1.903 | -4.096 | 3.564* | 0.892 | 1.934 |
|  | (0.84) | (-0.31) | (1.66) | (0.40) | (0.35) |
| dage | -0.000** | -0.002 | -0.000 | -0.000 | -0.001* |
|  | (-2.30) | (-0.80) | (-0.33) | (-0.52) | (-1.83) |
| dage2 | 0.000* | 0.000 | 0.000 | 0.000 | 0.000* |
|  | (1.72) | (0.60) | (0.13) | (0.27) | (1.72) |
| Constant | 5.763*** | 1.767*** | 2.981*** | 5.950*** | 3.487*** |
|  | (137.36) | (8.27) | (53.35) | (125.62) | (43.13) |
| Area FE | YES | YES | YES | YES | YES |
| Year FE | YES | YES | YES | YES | YES |
| Observations | 3,171 | 1,951 | 2,811 | 3,171 | 2,805 |
| R-squared | 0.907 | 0.319 | 0.751 | 0.886 | 0.676 |

revenues allocated specifically for supporting public utilities, and a decrease in transfers received. Consequently, the local government's capacity to facilitate infrastructure projects or assist firms has weakened, leading to a decline on firm-level TFP.

## 6.2 Impact of the TCD reform on the firms' internal environment

Changes in the external environment can directly impact firm performance [50]. Williamson [51] emphasizes the close relationship between government regulation and firms' operations. Some researchers have discovered that government fiscal policies can significantly affect firms' operational activities [52–54]. Total factor productivity (TFP) is a crucial indicator of firm performance and is naturally influenced by changes in the external environment. Aghion et al. [36] find that the tax burden on firms can affect the increase in TFP. Moreover, firms' investment in improving TFP is unpredictable and risky [55], creating higher demands on firms' operational and cash flow conditions. Therefore, the operational capacity and financing constraint of firms also exert significant impact on the firm-level TFP [56, 57].

As previously mentioned, the TCD reform has impacted the government's fiscal situation, resulting in alterations in transaction efficiencies. Furthermore, as authority shifts, there will be a reduction in local government revenue and therefore less incentive to promote economic development. For instance, the local government may decrease tax incentives and other supportive policies for businesses, implying that the TCD reforms have worsened the external environment for firm operations. Consequently, these changes in the external environment

**Table 9. Changes in the firms' internal situation.**

| | (1) | (2) | (3) | (4) | (5) |
|---|---|---|---|---|---|
| | **tax** | **addvalue** | **rota** | **fc** | **labor** |
| policy | 0.265*** | -0.046*** | -0.005** | 0.059*** | -0.018** |
| | (6.80) | (-3.44) | (-2.36) | (3.07) | (-2.10) |
| Constant | 2.411*** | 8.446*** | 0.092*** | -2.482*** | 4.608*** |
| | (81.82) | (797.80) | (25.97) | (-185.40) | (716.05) |
| Control variables | YES | YES | YES | YES | YES |
| Firm FE | YES | YES | YES | YES | YES |
| Year FE | YES | YES | YES | YES | YES |
| Industry FE | YES | YES | YES | YES | YES |
| Observations | 443,594 | 438,804 | 443,946 | 412,928 | 444,078 |
| R-squared | 0.674 | 0.854 | 0.639 | 0.756 | 0.914 |

could lead to adjustments in the internal circumstances of firms, including tax burden, return on assets, and financing constraints, ultimately affecting firm-level TFP.

Hence, we conduct a more in-depth analysis regarding the transformations in the financial and employment circumstances of local firms following the TCD reform. This is done by examining factors such as the tax burden(tax), industrial added value (addvalue), return on total assets (rota), financing constraints (fc), and the number of employees of the firms (labor). Columns (1)—(4) of Table 9 depict the decline in the internal financial condition of firms. Column (1) reveals a noteworthy rise in firms' tax burden subsequent to the reform. Columns (2) and (3) demonstrate a decrease in the industry value-added and the return on total assets for firms, indicating a decline in their profitability. Column (4) showcases a deterioration in firms' financing constraints, while column (5) also suggests a tendency for firms to reduce the number of employees following the reform. All these findings robustly indicate that the TCD reform has considerably worsened the internal operating environment of firms, consequently leading to a decline on firm-level total factor productivity.

## 6.3 Impact of the entry-exit behavior of firms

City-county mergers can help break down administrative barriers between regions, strengthen connections between different areas, and facilitate the flows of capital, labor, and other factors. However, because new municipal areas are often located on the outskirts of cities, prefecture governments may not focus as much on new municipal areas as they do on old ones. Additionally, the older municipal areas are often economically stronger than the new ones. As a result, after the reforms, the old municipal areas may siphon more production factors, exacerbating the "core-periphery" effect [12]. This could lead to high-productivity firms migrating to the older areas, while the new municipal areas only attract low-productivity firms. Finally, the decrease in firm-level TFP after the TCD reforms is probably owing to the entry-exit behavior of firms.

To test this mechanism, this paper separately builds the cross-multipliers of policy and firm-entry (*policy*\**entry*), and firm-exit (*policy*\**exit*), respectively. The results are shown in the Table 10. The coefficient of *policy*\**entry* in column (1) suggests that firm entry exacerbates policy effects, indicating that the entry of inefficient firms leads to a reduction in firm-level TFP in the reformed county-level cities. The coefficient of *policy*\**exit* in column (2) is statistically significant at 1% level, indicating that firms' exit behavior also amplifies the negative impact of TCD reforms on firm-level TFP. Additionally, we test the impact of the TCD reforms on non-

**Table 10. The impact of the entry-exit behavior of firms.**

|  | (1) | (2) | (3) | (4) |
|---|---|---|---|---|
|  | TFP | TFP | TFP | TFP |
| policy | -0.042*** | -0.024* | -0.042*** | -0.024* |
|  | (-3.53) | (-1.71) | (-3.51) | (-1.71) |
| policy*entry | -0.862*** |  |  |  |
|  | (-2.69) |  |  |  |
| policy*exit |  | -0.083*** |  |  |
|  |  | (-3.48) |  |  |
| Constant | 5.514*** | 5.498*** | 5.501*** | 5.499*** |
|  | (525.89) | (589.22) | (593.32) | (586.87) |
| Control variables | YES | YES | YES | YES |
| Firm FE | YES | YES | YES | YES |
| Year FE | YES | YES | YES | YES |
| Industry FE | YES | YES | YES | YES |
| Observations | 411,575 | 411,575 | 405,218 | 407,734 |
| R-squared | 0.793 | 0.793 | 0.793 | 0.793 |

entry and non-exit firms. Columns (3) and (4) show that the coefficients of policy remain significantly negative after excluding the effects of entry and exit firms, respectively, suggesting that the TCD reforms also have a negative effect on TFP for continuing firms.

## 7. Conclusion

After 1998, China shifted its focus on implementing city-county merger (also known as *che xian she qu*), wherein counties or county-level cities are transitioned as subservient municipal districts of prefecture cities. This process led to the centralization of certain administrative and financial powers from the reformed county-level governments to the prefecture-level city governments, resulting in changes in governments' decision-making within the reformed areas. The shift in decision-making and resource allocation due to the city-county merger may impact on firm performance, leading to variations in productivity.

To investigate the impact of city-county mergers on firm-level TFP, this paper presents a simple theoretical model to explain that the reduction in authority within local government leads to decreased "efficiency of transactions", ultimately resulting in a decline in firm-level productivity. After that, the policy of transitioning county-level cities into municipal districts (TCD) is treated as a quasi-natural experiment to examine the impact of power shifts between prefecture and county-level governments on firm-level total factor productivity (TFP) in China. The empirical findings indicate that (1) the TCD reform leads to a significant reduction in the TFP of existing firms; (2) in heterogeneous tests, the TCD reform has an insignificant impact on firms in the economically developed eastern region, but has a significant impact on firms in the relatively less developed central-western regions. Furthermore, the TFP of non-SOEs is significantly and negatively affected by the reform, while the TFP of SOEs is not significantly affected; (3) the mechanism analysis reveals that the TCD reform alters the government's financial situation and firms' internal operating environment, resulting in a decline in firm-level TFP.

Our study indicates that, although power centralization may contribute to the overall productivity of urban areas [17], it negatively impacts the firm-level TFP of the reformed areas. This suggests that a centralized power structure may hinder the efficiency of firms operating

within the system. Therefore, when implementing city-county merger reform, policymakers should seek a balance for efficient governance. Additionally, relevant powers should be appropriately redistributed to lower-level governments to maintain sufficient incentives. Ultimately, by optimizing power distribution within the political system, firms operating in the economic environment can better navigate regulations and policies, leading to improved performance. Therefore, while China's political system has its unique characteristics, the implications of this study extend beyond the borders of China, offering valuable insights for policymakers and leaders in other countries with similar political structures.

In addition, the insights gained from the experiences of the world's largest developing country have significant implications for other nations with emerging markets. We have observed that TCD reforms result in a reduction in the efficiency of institutional transactions, leading to a decrease in firm-level total factor productivity (TFP). Therefore, we suggest that developing countries undergoing economic transitions prioritize efforts to improve the effectiveness of intra-territorial transactions by expanding the provision of public services such as infrastructure. Moreover, our findings demonstrate that changes in the external policy environment can impact the internal operating environment of firms. In conclusion, policymakers should strive to maintain policy coherence and reduce policy uncertainty to mitigate the impact of policy changes on firm operations.

## Author Contributions

**Conceptualization:** Sisi Yin.

**Data curation:** Sisi Yin.

**Formal analysis:** Sisi Yin, Zuyi Shen.

**Funding acquisition:** Yunshu Gao.

**Writing – review & editing:** Yunshu Gao.

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
