## [Decision Letter · Decision Letter 0]

22 Feb 2024

PONE-D-24-00346The Impact of City-county Merger on Firm-level Total Factor Productivity —— Evidence from ChinaPLOS ONE

Dear Dr. YIN,

Thank you for submitting your manuscript to PLOS ONE. After careful consideration, we feel that it has merit but does not fully meet PLOS ONE’s publication criteria as it currently stands. Therefore, we invite you to submit a revised version of the manuscript that addresses the points raised during the review process. Both reviewers think that this is an interesting topic, and see the potential contribution of your article.  They have identified a number of areas of potential improvement, however.  These include:Exploring some alternative estimation approaches, and better justifying the approaches that you have selected.Checking and resolving inconsistencies between results in different tables or between text and tables (see Reviewer #1's comments)Clarifying the mechanisms through which the relationship between city-county mergers and the outcome variable.Broadening the scope of literature reviewed.

I encourage you to make revisions responsive to the feedback of both reviewers.

We look forward to receiving your revised manuscript.

Kind regards,

Joshua L Rosenbloom

Academic Editor

PLOS ONE

Journal Requirements:

4. We note that Figures 3 and 4 in your submission contain [map/satellite] images which may be copyrighted. All PLOS content is published under the Creative Commons Attribution License (CC BY 4.0), which means that the manuscript, images, and Supporting Information files will be freely available online, and any third party is permitted to access, download, copy, distribute, and use these materials in any way, even commercially, with proper attribution. For these reasons, we cannot publish previously copyrighted maps or satellite images created using proprietary data, such as Google software (Google Maps, Street View, and Earth). For more information, see our copyright guidelines: http://journals.plos.org/plosone/s/licenses-and-copyright.

a. You may seek permission from the original copyright holder of Figures 3 and 4 to publish the content specifically under the CC BY 4.0 license.  

Reviewers' comments:

Reviewer's Responses to Questions

**Comments to the Author**

1. Is the manuscript technically sound, and do the data support the conclusions?

Reviewer #1: Yes

Reviewer #2: Yes

2. Has the statistical analysis been performed appropriately and rigorously? 

Reviewer #1: Yes

Reviewer #2: Yes

3. Have the authors made all data underlying the findings in their manuscript fully available?

Reviewer #1: No

Reviewer #2: No

4. Is the manuscript presented in an intelligible fashion and written in standard English?

Reviewer #1: Yes

Reviewer #2: Yes

5. Review Comments to the Author

Reviewer #1: The paper examines the impact of city-county mergers, mainly the transition of county-level cities

into municipal districts (TCD) reform, on firm-level total factor productivity (TFP) in China. In this paper, the authors establish a simple theoretical model and employ the Differences-in-Differences (DIDs) method to explain that the reduction in authority within local government leads to a decreased “efficiency of transactions”, resulting in a decline in firm-level productivity.

The paper addresses a timely issue in China and manages to explore possible explanations for it. However, I have some concerns in term of regression details and the interpretation of the results, listed as follows:

1. In the robust test (in section 5.2.4), this paper exclusively employs the CSDID method to tackle the heterogeneous treatment effects of the Two-Way Fixed Effects (TWFE) estimator in the baseline under static conditions. However, it is equally crucial to derive the firm-year average treatment effect under dynamic conditions, as shown in Figure 5. Moreover, there have been several proposed "heterogeneous-robust" estimators that warrant further verification.

2. The sample numbers for some columns of regression were inconsistent due to missing value of control variables or other variables, such as in table 2, columns (1)-(4); table 3, columns (1)-(4); table 4, columns (1)-(4); table 6, column (6); table 8, columns (1)-(5); table 9, columns (1)-(5).

3. In table 6, according to the author's explanation, policy 1 (strengthen counties by expanding their powers) and policy 2 (province-managing county) both enhanced the administrative power of the county. However, the coefficients for policy 1 unexpectedly appear negative in columns (3) and (4), whereas the coefficients for policy 2 are positive in both columns (3) and (4).

4. The mechanism analysis through which the TCD reform can impact firm-level TFP is not clear and direct. According to the authors’ explanation, the TCD reform has had an impact on the government’s fiscal situation (checked in table 8), leading to changes in (institutional and non-institutional) transaction efficiencies (explained in section 2). Moreover, the changes in transaction efficiency from city-county mergers can influence firm performance (checked in table 9), leading to variations in productivity. However, the inframarginal model developed by Yang & Shi (1992) only explores how the changes in transaction efficiency from county-city mergers affect firm-level TFP. I wonder what role the model plays in the mechanism of firms’ internal environment, and there is a lack of a detailed explanation of how adjustments in the external environment might trigger changes in the internal circumstances of firms.

Regarding the mechanism of negative effects, another possible explanation is that the city-county mergers have strengthened the administrative jurisdiction of the municipal government over the counties, reduced administrative barriers, promoted the smoother flow of various resource elements such as capital and labor, and affected the entry and exit behavior of enterprises. In other words, it has led to the entry of some less efficient enterprises in a short period of time, or the exit of more efficient enterprises.

Other Comments:

1. Typos. In section 6.2, please note the absence of parentheses in several instances. Change “Column 1” to “Column (1)” for consistency, as well as “Column 2”, ..., and “Column 5”, etc."

Reviewer #2: The paper analyzes the effect of city county merger on firm productivity in the context of China. This is an interesting piece of work, but it needs further revision to meet the standards of Plos One.

1. Estimation - Authors use a DD regression framework for the analysis and performs standard robustness checks. However, what is the level of clustering of standard errors for the purpose of the analysis is not clear. The standard errors should be clustered at the city level. Further, it is advised to use wild bootstrapped standard errors for inference.

2. Authors show heterogeneity based on region and ownership (SOE and non-SOE). However, what about heterogeneity based on age and size of the firm? This assumes particular importance if financing constraints is one of the channels driving the result. In such a scenario ,I would expect to observe higher effects for small/ financially constrained firms.

3. Authors largely cite existing literature from China. I suggest authors should also highlight other studies in different country settings and engage with greater research diaspora.

4. The implications of these findings for emerging market economies should be communicated.

6. PLOS authors have the option to publish the peer review history of their article (what does this mean?). If published, this will include your full peer review and any attached files.

Reviewer #1: No

Reviewer #2: No

---

## [Author Response · Author response to Decision Letter 0]

17 Apr 2024

Dear reviewer #1 and #2,

 The author has carefully revised the suggested modifications from the reviewers, please see attached files (response to review #1 and reponse to review #2) for details.

Yours Sincerely,

Sisi Yin

---

## [Decision Letter · Decision Letter 1]

24 Apr 2024

PONE-D-24-00346R1The Impact of City-county Merger on Firm-level Total Factor Productivity —— Evidence from ChinaPLOS ONE

Dear Dr. YIN,

Thank you for submitting your manuscript to PLOS ONE. After careful consideration, we feel that it has merit but does not fully meet PLOS ONE’s publication criteria as it currently stands. Therefore, we invite you to submit a revised version of the manuscript that addresses the points raised during the review process.

**You have successfully addressed the concerns raised by the reviewers regarding the earlier version of your manuscript.  However, the final paragraph of the conclusion, which you have added, is not (as reviewer #2 observes) well linked to the analysis you have conducted. ** 

We look forward to receiving your revised manuscript.

Kind regards,

Joshua L Rosenbloom

Academic Editor

PLOS ONE

Journal Requirements:

Reviewers' comments:

Reviewer's Responses to Questions

**Comments to the Author**

1. If the authors have adequately addressed your comments raised in a previous round of review and you feel that this manuscript is now acceptable for publication, you may indicate that here to bypass the “Comments to the Author” section, enter your conflict of interest statement in the “Confidential to Editor” section, and submit your "Accept" recommendation.

Reviewer #1: All comments have been addressed

Reviewer #2: All comments have been addressed

2. Is the manuscript technically sound, and do the data support the conclusions?

Reviewer #1: Yes

Reviewer #2: Yes

3. Has the statistical analysis been performed appropriately and rigorously? 

Reviewer #1: Yes

Reviewer #2: Yes

4. Have the authors made all data underlying the findings in their manuscript fully available?

Reviewer #1: Yes

Reviewer #2: (No Response)

5. Is the manuscript presented in an intelligible fashion and written in standard English?

Reviewer #1: Yes

Reviewer #2: Yes

6. Review Comments to the Author

**Reviewer #1:** (No Response)

**Reviewer #2:** Authors' have substantially revised the manuscript in light of the comments. However, the last paragraph of the paper seems to far fetched and the implications should be linked to the results. For example consider the following:

"In such scenarios, to bolster productivity and foster sustained economic growth, it is recommended that developing countries in the midst of economic transitions focus on enhancing the trade environment, improving the efficiency of intra-territorial transactions, and reducing firms' transaction costs. Moreover, providing government incentives to firms are also crucial."

How is the above flowing from the analysis?

Authors should revise the conclusion section and make it coherent.

7. PLOS authors have the option to publish the peer review history of their article (what does this mean?). If published, this will include your full peer review and any attached files.

Reviewer #1: No

Reviewer #2: No

---

## [Author Response · Author response to Decision Letter 1]

4 May 2024

Dear Reviewers,

The detailed revisions are attached to the files.

Best regards,

Sisi Yin

---

## [Editor Report · Decision Letter 2]

7 May 2024

The Impact of City-county Merger on Firm-level Total Factor Productivity —— Evidence from China

PONE-D-24-00346R2

Dear Dr. YIN,

We’re pleased to inform you that your manuscript has been judged scientifically suitable for publication and will be formally accepted for publication once it meets all outstanding technical requirements.

Kind regards,

Joshua L Rosenbloom

Academic Editor

PLOS ONE
---

## [Editor Report · Acceptance letter]

8 Jul 2024

PONE-D-24-00346R2 

PLOS ONE

Dear Dr. YIN, 

I'm pleased to inform you that your manuscript has been deemed suitable for publication in PLOS ONE. Congratulations! Your manuscript is now being handed over to our production team.

Kind regards, 

on behalf of

Dr. Joshua L Rosenbloom 

Academic Editor

PLOS ONE